# Elicitation of Broadly Neutralizing Antibodies against B.1.1.7, B.1.351, and B.1.617.1 SARS-CoV-2 Variants by Three Prototype Strain-Derived Recombinant Protein Vaccines

**DOI:** 10.3390/v13081421

**Published:** 2021-07-22

**Authors:** Yong Yang, Jinkai Zang, Shiqi Xu, Xueyang Zhang, Sule Yuan, Haikun Wang, Dimitri Lavillette, Chao Zhang, Zhong Huang

**Affiliations:** CAS Key Laboratory of Molecular Virology & Immunology, Institut Pasteur of Shanghai, Chinese Academy of Sciences, University of Chinese Academy of Sciences, Shanghai 200031, China; yyang@ips.ac.cn (Y.Y.); jkzang@ips.ac.cn (J.Z.); sqxu@ips.ac.cn (S.X.); xyzhang@ips.ac.cn (X.Z.); slyuan1@ips.ac.cn (S.Y.); hkwang@ips.ac.cn (H.W.); dlaville@ips.ac.cn (D.L.)

**Keywords:** COVID-19, SARS-CoV-2 variants, receptor binding domain, S1, spike protein, vaccine, neutralizing antibody

## Abstract

The ongoing coronavirus disease 2019 (COVID-19) pandemic is caused by severe acute respiratory syndrome coronavirus 2 (SARS-CoV-2). Most of the currently approved SARS-CoV-2 vaccines use the prototype strain-derived spike (S) protein or its receptor-binding domain (RBD) as the vaccine antigen. The emergence of several novel SARS-CoV-2 variants has raised concerns about potential immune escape. In this study, we performed an immunogenicity comparison of prototype strain-derived RBD, S1, and S ectodomain trimer (S-trimer) antigens and evaluated their induction of neutralizing antibodies against three circulating SARS-CoV-2 variants, including B.1.1.7, B.1.351, and B.1.617.1. We found that, at the same antigen dose, the RBD and S-trimer vaccines were more potent than the S1 vaccine in eliciting long-lasting, high-titer broadly neutralizing antibodies in mice. The RBD immune sera remained highly effective against the B.1.1.7, B.1.351, and B.1.617.1 variants despite the corresponding neutralizing titers decreasing by 1.2-, 2.8-, and 3.5-fold relative to that against the wild-type strain. Significantly, the S-trimer immune sera exhibited comparable neutralization potency (less than twofold variation in neutralizing GMTs) towards the prototype strain and all three variants tested. These findings provide valuable information for further development of recombinant protein-based SARS-CoV-2 vaccines and support the continued use of currently approved SARS-CoV-2 vaccines in the regions/countries where variant viruses circulate.

## 1. Introduction

Severe acute respiratory syndrome coronavirus 2 (SARS-CoV-2) is the causative agent of the ongoing coronavirus disease 2019 (COVID-19) pandemic [1,2,3]. The spike (S) protein of SARS-CoV-2 mediates virus entry into target cells and can be divided into the receptor-binding subunit S1, the membrane-fusion subunit S2, a transmembrane domain, and a short intracellular tail. The S1 subunit can be further divided into the N-terminal domain (NTD) and the C-terminal domain (CTD); the latter can directly bind to the host cell receptor angiotensin-converting enzyme 2 (ACE2) and is also referred to as the receptor-binding domain (RBD) [4,5]. The S, S1, and RBD proteins are major targets for SARS-CoV-2 vaccine development [6]. Thus far, multiple SARS-CoV-2 vaccines have been developed and widely implemented since December 2020, including mRNA vaccines, inactivated whole-virus vaccines, adenovector vaccines, and recombinant protein vaccines.

Since the discovery of SARS-CoV-2 in late 2019, the virus continues to evolve over time. Several SARS-CoV-2 variants of concern have emerged and are now circulating worldwide, including the B.1.1.7 lineage first identified in the United Kingdom, the B.1.351 lineage in South Africa, and the P.1 lineage in Brazil. These variants, especially B.1.351, contain multiple mutations in the S protein and show increased resistance to neutralization by some potent monoclonal antibodies, convalescent plasma, and vaccinee sera [6,7], thus threatening the protective efficacy of currently approved vaccines that were developed based on the original SARS-CoV-2 virus from initial outbreaks. Recently, a new SARS-CoV-2 variant termed B.1.617 emerged in India. The emergence of B.1.617 has been suggested to contribute to the current surge of SARS-CoV-2 infections in India [8]. It remains to be determined whether currently approved vaccines are effective against B.1.617 variants. In an attempt to assess the recombinant protein vaccines developed based on the SARS-CoV-2 prototype strain (Wuhan-Hu-1), three recombinant proteins, namely, RBD, S1, and S ectodomain trimer (S-trimer), were produced in mammalian cells and subsequently analyzed for their immunogenicity and ability to induce cross-neutralizing antibodies against SARS-CoV-2 variants. 

## 2. Materials and Methods

### 2.1. Cells

HEK 293F (Thermo Fisher, Waltham, MA, USA) suspension cells were cultured in FreeStyle 293 expression medium (Gibco, Grand Island, NY, USA). HEK 293T over-expressing human ACE2 (293T-hACE2) was generated in a previous study [9].

### 2.2. Protein Preparation

RBD, S1, and S-trimer derived from the SARS-CoV-2 prototype strain Wuhan-Hu-1 (GenBank ID: MN908947.3) were produced using the HEK 293F expression system with the Union 293 medium (Union-Biotech, Shanghai, China). Schematic diagrams of recombinant expression vectors are shown in Appendix A. Briefly, to generate RBD, the codon-optimized RBD gene fragment (residues V320 to G550) was cloned into pcDNA3.4 vector with N-terminal interleukin-10 (IL-10) signal sequence, Strep-tag II, and C-terminal His tag, yielding plasmid pcDNA3.4-RBD. To generate S1, an optimized S1 gene (residues V16 to R685) was cloned into pcDNA3.4 vector with N-terminal IL-10 signal sequence and C-terminal His tag, yielding plasmid pcDNA3.4-S1. To prepare S-trimer, an optimized S gene (residues M1 to Q1208) was cloned into the pcDNA3.4 vector with a C-terminal T4 fibritin trimerization motif, a human rhinovirus 3C protease cleavage site, a Twin-Strep-tag, and a His tag. To stabilize the S-trimer protein, “GSAS” substitution at the furin S1/S2 cleavage site (residues 682–685) and proline mutations at residues 986 and 987 [10] were introduced into the S gene of this plasmid using the NEBuilder HiFi DNA Assembly Master Mix (NEB, Ipswich, MA, USA), resulting in the plasmid pcDNA3.4-S-trimer. Each plasmid was transfected into HEK 293F cells, and His-tagged proteins in the culture supernatants were purified using Ni-NTA resin (EMD Millipore, Darmstadt, Germany) according to manufacturer’s protocol. The purified proteins were quantified by Bradford assay and then analyzed by SDS-PAGE and western blotting, as described below.

### 2.3. SDS-PAGE and Western Blotting

Purified RBD, S1, and S-trimer protein samples were separated on 12% SDS-PAGE gels. A pre-stained protein marker (10kDa~250kDa) (EpiZyme, Shanghai, China) was used. After electrophoresis, the gels were stained with Coomassie blue R-250 or transferred onto polyvinylidene difluoride (PVDF) membranes (Pall, Port Washington, NY, USA). For immunodetection, the PVDF membranes were incubated with an RBD-specific polyclonal antibody generated in-house, followed by horseradish peroxidase (HRP)-conjugated goat anti-mouse IgG (Sigma-Aldrich, Saint Louis, MO, USA).

### 2.4. Mouse Immunization

The animal studies were approved by the Institutional Animal Care and Use Committee at the Institut Pasteur of Shanghai (A2020008, 6 February 2020).

Groups of six female BALB/c mice (6–8 weeks old) were immunized intraperitoneally (i.p.) with wild-type RBD, S1, or S-trimer proteins (10 μg/dose) or PBS in combination with 0.5 mg aluminum hydroxide adjuvant (Invivogen, San Diego, CA, USA) at weeks 0, 6, and 12. Blood samples were taken from each mouse at weeks 18 and 32 (Figure 1B). Antiserum samples were inactivated at 56 °C for 30 min before analyses.

### 2.5. ELISA

To determine the S-specific serum antibody titer, ELISA plates were coated with 50 ng/well of the wild-type S-trimer at 4 °C overnight. After blocking with 5% skim milk in PBS with 0.05% Tween 20 (PBST) and washing, the plates were incubated with 50 μL of twofold serially diluted antisera for 1.5 h at 37 °C. After washing, 50 μL of HRP-conjugated anti-mouse IgG (Sigma-Aldrich; diluted 1:10,000 in 1% milk/PBST) was added and incubated for 1 h. After washing and color development, absorbance was measured at 450 nm.

### 2.6. Receptor Competition ELISA

An ACE2 competition ELISA assay was carried out as described previously with some modifications [11]. Briefly, ELISA plates were coated with 50 ng/well of wild-type S-trimer, followed by blocking with 5% milk in PBST. The plates were then incubated with the mixtures of biotinylated hACE2-Fc (20 ng/well) and serially diluted antisera, followed by incubation with HRP-conjugated streptavidin (Life Technologies, Grand Island, NY, USA).

### 2.7. Pseudovirus Neutralization Assay

Murine leukemia virus (MLV)-based SARS-CoV-2 S pseudoviruses were generated according to our previously published protocol [9], with the exception that plasmids encoding full-length S protein of SARS-CoV-2 B.1.1.7 [12], B.1.351 [13], or B.1.617.1 [14] variants (Appendix A) were also used in the current study. The mutant plasmids were constructed using the Mut Express^TM^ II Fast Mutagenesis Kit V2 (Vazyme, Nanjing, China) following the manufacturer’s protocol.

For the neutralization assay, three-fold serial dilutions of serum samples (50 μL/well) were mixed with equal volumes of pseudovirus and incubated for 1 h at 37°C. The mixtures were added into 96-well plates, into which 293T-hACE2 cells had been seeded for 20 h. After incubation for 12 h, the sera/pseudovirus mixtures were removed and fresh DMEM containing 2% FBS was added to the wells. After 48 h, the cells were lysed with lysis buffer (Promega, Madison, WI, USA), and luciferase activity was measured using the luciferase assay system (Promega). The 50% neutralization titer (NT50) was calculated by GraphPad Prism software (version 7.0, San Diego, CA, USA).

### 2.8. Statistical Analysis

All statistical analyses were performed with GraphPad Prism software. Statistical significance was analyzed using Student’s *t*-test.

## 3. Results

### 3.1. Production of Recombinant Protein Antigens

Three expression vectors, namely, pcDNA3.4-RBD, pcDNA3.4-S1, and pcDNA3.4-S-trimer, were constructed and used to transfect HEK 293F cells (Appendix A). Note that the S-trimer protein was stabilized in the pre-fusion conformation by the proline substitution and C-terminal foldon trimerization motif [10]. Recombinant proteins were purified from culture supernatants of the plasmids-transfected HEK 293F cells. The identity of the three recombinant proteins was confirmed by SDS-PAGE and western blot analyses (Figure 1A).

### 3.2. Immunogenicity Analysis

For immunogenicity comparison, groups of six BALB/c mice were immunized with alum-formulated RBD, S1, or S-trimer proteins (10 μg/dose) or PBS at weeks 0, 6, and 12, and serum samples were taken from each mouse at weeks 18 and 32 (Figure 1B). Individual antisera were measured for antigen-specific IgG titers using ELISA with S-trimer protein as the coating antigen. The results are shown in Figure 1C. As expected, control sera from mice immunized with PBS/alum did not show any S-binding activity. In contrast, all week-18 sera of the RBD, S1, and S-trimer groups reacted strongly with the S-trimer antigen with geometric mean titers (GMT) of 320,000, 113,137, and 320,000, respectively. Notably, the S-binding titers of the RBD and the S-trimer groups were significantly higher than those of the S1 group (*p* < 0.05), suggesting that RBD and S-trimer are more immunogenic than S1. At week 32, the S-binding titers of the antisera from the three vaccine groups maintained high levels (*p* > 0.05) comparable to those of the corresponding week-18 antisera. The GMTs for the week-32 anti-RBD, anti-S1, and anti-S-trimer sera were 285,088, 71,272, and 126,992, respectively. These data show that the three candidate vaccines, especially RBD and S-trimer, can induce long-lasting antigen-specific antibody responses in the immunized mice. 

We performed receptor competition ELISA to assess the function of the vaccine-induced sera. Briefly, the week-18 antisera were allowed to compete with the soluble hACE2 receptor for binding to S-trimer coated on the ELISA plates and their blocking efficiency was determined by measuring the decrease in hACE2-binding signals. As shown in Appendix A, anti-RBD, anti-S1, and anti-S-trimer sera could effectively inhibit S/hACE2 binding with 50% blocking titers (GMTs) of 6930, 1256, and 3253, respectively, whereas the control sera did not exhibit any blockade activity even at the 1:100 dilution (the lowest dilution tested). In addition, the blocking titers of the antisera appeared to correlate with the S-binding titers of the corresponding antisera (Appendix A). These data suggest that the vaccine-induced antisera can potentially prevent viral entry by blocking the binding of S protein to the cellular hACE2 receptor. 

We then performed pseudovirus neutralization assays to determine the neutralization capacity of the vaccine-elicited antisera. The week-18 antisera were first tested for neutralization of pseudovirus bearing wild-type (Wuhan-Hu-1 strain) SARS-CoV-2 S protein, and the results are shown in Figure 1D. All week-18 sera of the RBD, S1, and S-trimer groups were able to potently neutralize pseudovirus, whereas antisera from the control (PBS) group did not show any neutralization activity. The 50% pseudovirus neutralization titers (pVNT50) of the RBD and S-trimer groups were 23,393 and 16,068, respectively, significantly higher than that of the S1 group (4633). The antisera collected at week 32 (20 weeks after the last booster) were then tested for neutralization of the wild-type pseudovirus. For the RBD and S1 vaccine groups, the neutralization titers of the week-32 sera were comparable to those of the corresponding week-18 sera (*p* < 0.05). For the S-trimer group, the neutralization potency of the week-32 sera significantly decreased compared to that of the week-18 sera but remained at a high level (geometric mean pVNT50 = 6956). Nonetheless, these data indicate that the three subunit vaccines can induce long-lasting neutralizing antibodies in mice.

### 3.3. Neutralization against SARS-CoV-2 Variants

To assess the neutralization breadth of the antisera induced by the prototype vaccine antigens, we generated a panel of pseudoviruses representing SARS-CoV-2 variants, including B.1.1.7, B.1.351, and B.1.617.1 (Appendix A). The week-18 sera were tested for their cross-neutralization abilities against the pseudovirus panel, and the results are shown in Figure 1E. Anti-RBD sera potently neutralized the wild-type (Wuhan-Hu-1 strain) SARS-CoV-2 pseudovirus (GMT = 23,393); however, neutralizing titers of anti-RBD sera against B.1.1.7, B.1.351, and B.1.617.1 were 1.2-, 2.8-, and 3.5-fold lower than that against the wild-type, respectively (GMT = 19,243, 8453, and 6695, respectively). A similar neutralization pattern was observed for S1 immune sera; specifically, neutralizing GMTs of anti-S1 sera against B.1.1.7, B.1.351, and B.1.617.1 were 2448, 2147, and 1233, respectively. For the S-trimer immune sera, the neutralizing antibody titers against the B.1.1.7, B.1.351, or B.1.617.1 variants were comparable (*p* < 0.05) to those against the wild-type pseudovirus. 

## 4. Discussion

In the present study, we compared three recombinant protein SARS-CoV-2 vaccines developed based on a single prototype strain for their ability to induce broadly neutralizing antibodies against the wild-type strain and major circulating variants. The results showed that, at the same antigen dose, the RBD and S-trimer vaccines were more potent than the S1 vaccine in eliciting long-lasting, high-titer broadly neutralizing antibodies in mice. The RBD immune sera remained highly effective against the B.1.351 and B.1.617.1 variants despite the corresponding neutralizing titers decreasing by 2.8- to 3.5-fold relative to that against the wild-type strain. Significantly, the S-trimer immune sera exhibited comparable neutralization potency (less than twofold variation in neutralizing GMTs) towards the prototype strain and all three variants tested. Compared to the RBD- and S-trimer-immunized mouse sera, the S1 immune sera showed inferior neutralizing activity towards the three variants. Possible explanations for this phenomenon include the following: (1) S1 protein is less immunogenic than RBD and S-trimer, as indicated by the lower levels of antibody responses induced by S1 (Figure 1C); (2) the deletions and mutations in the NTD region of the variants may abrogate neutralization by most NTD-targeted neutralizing antibodies [15]. 

In this study, we adopted a three-dose vaccination strategy. Although not compared directly in this study, it is likely that a three-dose regimen will provide higher neutralizing antibody titers and more effective protection than a two-dose regimen. Actually, it has been shown in a previous study that, for recombinant protein-based SARS-CoV-2 vaccines, a third dose can confer a strong boost to antibody response [16]. In fact, the recombinant dimeric RBD protein-based SARS-CoV-2 vaccine (ZF2001), which has been approved for human use in China, is administered in three doses. Significantly, sera from the ZF2001 vaccinees showed potent neutralizing activity against the B.1.351 variant [17]. Hence, the data from the present and previous studies demonstrate that SARS-CoV-2 prototype stain-derived recombinant protein vaccines can potently induce cross-variant neutralizing antibodies. These findings provide valuable information for further development of recombinant protein-based SARS-CoV-2 vaccines and support the continued use of currently approved SARS-CoV-2 vaccines (e.g., ZF2001) in the regions/countries where variant viruses circulate.

## Figures and Tables

**Figure 1 viruses-13-01421-f001:**
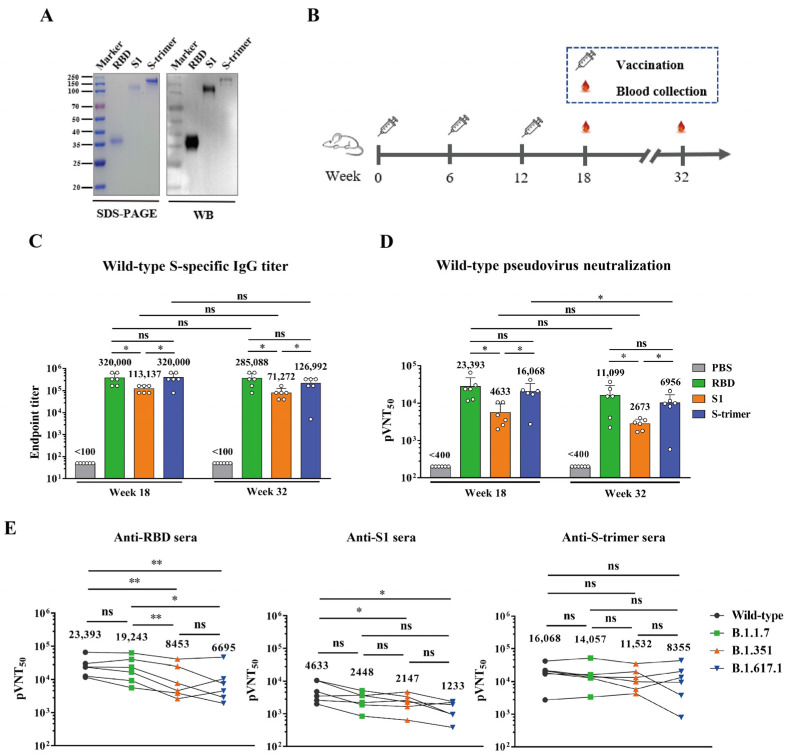
Induction of broadly neutralizing antibodies against SARS-CoV-2 variants by recombinant RBD, S1, and S-trimer vaccines in mice. (**A**) SDS-PAGE and western blot (WB) analysis of purified RBD, S1, S-trimer antigens. An anti-RBD (inclusion bodies) polyclonal antibody was used as the detection antibody; (**B**) Mouse immunization and sampling schedule. Four groups of mice were immunized three times with either RBD/alum, S1/alum, S-trimer/alum, or PBS/alum; (**C**) S protein-specific serum antibody titers were determined by ELISA. Anti-PBS sera did not exhibit any binding at the lowest serum dilution (1:100) and was assigned a titer of 50 for calculation; (**D**) Neutralizing antibody titers of the antisera against wild-type SARS-CoV-2 pseudovirus. Anti-PBS sera did not exhibit any neutralization activity at the lowest serum dilution (1:400) and was assigned a titer of 200 for calculation. PVNT50, 50% pseudovirus neutralization titer. For panels (**C**,**D**), serum samples collected at weeks 18 and 32 were used; (**E**) Neutralizing antibody titers of the week-18 antisera against SARS-CoV-2 prototype, B.1.1.7, B.1.351, and B.1.617.1 pseudoviruses. Data of a given antiserum sample were linked to trace its neutralization titers against different pseudoviruses. For panels (**C**–**E**), each symbol represents one mouse. Geometric mean was calculated for each set of data, shown and compared. Statistical significance was determined by Student’s t-test and is indicated as follows: ns, not significant, * *p* < 0.05, ** *p* < 0.01. Error bars represent SD.

## Data Availability

All data are available from the corresponding authors upon reasonable request.

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
