# Peer review of "Elicitation of Broadly Neutralizing Antibodies against B.1.1.7, B.1.351, and B.1.617.1 SARS-CoV-2 Variants by Three Prototype Strain-Derived Recombinant Protein Vaccines"

_viruses, 2021, doi:10.3390/v13081421_

Round 1

Reviewer 1 Report

The study entitled “Elicitation of broadly neutralizing antibodies against B.1.1.7, B.1.351, and B.1.617.1 SARS-CoV-2 variants by three prototype strain-derived recombinant protein vaccines” by Yong Yang et al., is an interesting well-written study. Minor comments as follow:

  • Line 51 please correct “current approved” to “currently approved”
  • Line 75: “By contrast “ to “In contrast “
  • Line 94 please correct “to cellular hACE2 receptor” to “to the cellular hACE2 receptor”
  • Line 128 please remove “More information was provided in” it is enough to state Supplementary Figure. S3).
  • Lines 130-131: any explanation of that the anti-RBD neutralized B.1.1.7 variant with higher GMT titer (31,581) than the wild type Wuhan-Hu-1 strain of which the plasmids were developed)
  • Lines 142-14: although this short communication, but authors need to discuss why the S1 vaccine is of inferior neutralizing activity against the variant SARS-COV-2 and to highlight the relation of their findings to the currently approved vaccine
  • Line 187: the authors need to address if more than 2 doses are recommended to the current vaccines to be able to neutralize the new variants? As I see the study is based on 3 vaccination doses at 0, 6, and 12 weeks
  • Line 149-150: the statement “and also support the continued use of current approved SARS-CoV-2 vaccines in the regions/countries where variant viruses circulate”, which vaccines, are all vaccines are effective against the new variants, please clarify.

Author Response

We thank the reviewer for the positive overall evaluation. Our response are attached in the pdf file.

Reviewer 2 Report

The communication is written to a high standard and the data presented is of high quality.  I have a few minor comments relating to the text presented.  My only major comment is that I do not understand why this work has not been presented as a full research article and the supplementary files included within the body of the article; however, this is a choice for the authors and editor.

Minor comments.

  1. Figure 1, section E.  Why are the data points joined?  Perhaps the authors should briefly explain why in the figure legend.
  2. There is very little Discussion of the actual results which probably reflects the fact that this is a brief communication.  A couple of additional sentences might help frame the context of the findings but this is a choice for the authors/editor.

Author Response

(The authors gave the same response as above.)
